

# Rapid rise in toxic load for bees revealed by analysis of pesticide use in Great Britain

Dave Goulson, Jack Thompson and Amy Croombs

School of Life Sciences, University of Sussex, Falmer, East Sussex, UK

## ABSTRACT

A strong argument can be made that the European Union has the most rigorous regulatory system for pesticides in the world, and that modern pesticide use poses fewer environmental threats than older regimes. Nevertheless, the impacts of pesticides on bees and other non-target organisms are much debated in Europe as elsewhere. Here we document changing patterns of pesticide use in arable and horticultural crops in Great Britain from 1990 to 2015. The weight of pesticides used has approximately halved over this period, but in contrast the number of applications per field nearly doubled. The total *potential* kill of honeybees (the total number of $LD_{50}$ doses applied to the 4.6 million hectares of arable farmland in Great Britain each year) increased six-fold to approximately $3 \times 10^{16}$ bees, the result of the increasing use of neonicotinoids from 1994 onwards which more than offset the effect of declining organophosphate use. It is important to stress that this does not mean that this number of bees will be killed, and also to acknowledge that our simple analysis does not take into account many factors such as differences in persistence, and timing and mode of application of pesticides, which will affect actual exposure of non-target organisms. Nonetheless, all else being equal, these data suggest that the risk posed by pesticides to non-target insects such as bees, other pollinators and natural enemies of pests, has increased considerably in the last 26 years.

**Subject** Ecotoxicology
**Keywords** Insecticide, Neonicotinoid, *Apis mellifera*, Pyrethroid, Organophosphate, Toxicity, $LD_{50}$

## INTRODUCTION

There is widespread concern regarding the health of populations of insect pollinators including domestic honey bees (*Apis mellifera*) and wild pollinators such as bumblebees (*Bombus* sp.). There is clear evidence for significant declines in the abundance and distribution of many pollinators, with some local and global extinctions (reviewed in *Goulson et al., 2015*). This has given rise to concerns over the future supply of pollination services for crops, currently valued at about €153 billion globally (*Gallai et al., 2009*). There is a broad consensus that these declines are due to a combination of factors including habitat loss, emergent diseases, exposure to pesticides, and climate change, although the relative importance of these factors is debated (*Godfray et al., 2015*; *Goulson et al., 2015*). The role of pesticides is the most controversial, since the debate impinges directly on farmers and the crop production industry.

In Europe, the regulatory system for pesticides is widely regarded as the most rigorous in the world, with a complex system in place to review the safety of new plant

Corresponding author
Dave Goulson,
d.goulson@sussex.ac.uk

**How to cite this article** Goulson et al. (2018), Rapid rise in toxic load for bees revealed by analysis of pesticide use in Great Britain.
PeerJ 6:e5255; DOI 10.7717/peerj.5255

protection products and re-evaluate their safety at intervals (*Handford, Elliot & Campbell, 2015*). Although pesticides are known to have wrought considerable environmental damage in the past, as was famously exposed by Rachel Carson's book "Silent Spring" (*Carson, 1962*), there is a perception that modern pesticides are much safer (*Dudley et al., 2017*). The European Union (EU) has been promoting reduced pesticide use and increased adoption of Integrated Pest Management (IPM) practices (*Hillocks, 2012*; *Handford, Elliot & Campbell, 2015*). Notably, the EU introduced a moratorium in 2013 which prevents the use of some neonicotinoid insecticides on flowering crops, a measure specifically intended to reduce risks to bees. Use of these chemicals elsewhere in the world is much less restricted. One might thus expect the EU to be a relatively benign region for bees, at least with regard to exposure to pesticides.

Nonetheless there are concerns that the landscape scale, industrial use of multiple pesticides poses risks to the environment that are not captured by regulatory tests which largely focus on short-term studies in which test organisms are exposed to a single chemical (*Dudley et al., 2017*; *Milner & Boyd, 2017*). Recent studies have linked environmental contamination with neonicotinoid insecticides in particular to declines in wild bees (*Woodcock et al., 2016*), aquatic insects (*Van Dijk, Van Staalduinen & Van der Sluijs, 2013*), butterflies (*Gilburn et al., 2015*; *Forister et al., 2016*) and insect-eating birds (*Hallmann et al., 2014*) (reviews in *Goulson, 2013*; *Pisa et al., 2015*; *Wood & Goulson, 2017*). All but one of these studies (*Forister et al., 2016*) were conducted in Europe.

Here we use a novel approach to evaluate whether the risks to bees posed by pesticide use in farming are decreasing or increasing, focussing on Great Britain for the simple reason that detailed pesticide use data are available for this region from 1990 to the present. We would expect similar patterns elsewhere, particularly across the EU (of which Great Britain remains part for the moment). We examine patterns of change in the mass of pesticides used, the area sprayed, and the total number of honey bees that could potentially be killed, in the period 1990–2015. Pesticide usage data was obtained from the Food and Environment Research Agency (*Defra, 2018*). We then calculated the number of honey bee $LD_{50}$ doses applied each year for each chemical, by dividing the mass applied by the $LD_{50}$ (this could be repeated for any organism for which $LD_{50}$ values are widely available). This indicates the *potential* kill due to use of the chemical, assuming all of it was consumed by or came into contact with honeybees. Clearly in the real world only a tiny fraction of pesticides used actually come into contact with non-target organisms. Our intention is not to quantify actual bee kills, but to provide a comparative measure of how the toxic load entering the environment has changed over 26 years. Although we focus on bees, our analysis is relevant to all insects, including those providing other important ecosystem services such as pest control and nutrient cycling.

## METHODS

Pesticide usage data were obtained from the Food and Environment Research Agency website (*Defra, 2018*). All 416 pesticides on the Defra database were initially included, but those for which total usage over the 1990–2015 survey period was below 100 kg were subsequently discarded, leaving 396 chemicals. Pesticide usage is recorded as both the mass

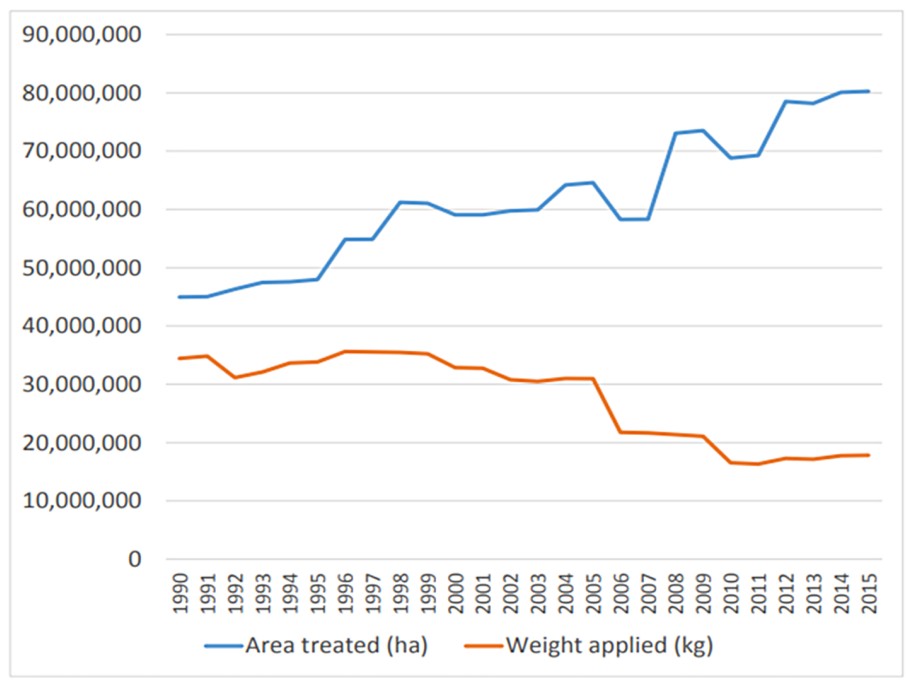

**Figure 1 Area of crop treated (blue line, hectares) and mass of pesticide applied (red line, kilograms) from 1990 to 2015.** The total area of crop remained approximately constant at 4.6 million hectares. In 1990 each hectare of cropped land on average received a total of 7.5 kg of pesticide active ingredient delivered in 9.8 applications. By 2015 each hectare of land received 3.9 kg of pesticide in 17.4 applications.

applied each year, and the area treated. For the latter, a treatment of 10 ha with one application of a product, or the treatment of one ha with 10 applications of a product in a year would both give a value of 10 ha treated.

We obtained $LD_{50}$ data for honey bees for each chemical from the existing literature. Wherever available, we used 48 h $LD_{50}$ values (oral or contact $LD_{50}$ values), but in a few cases these were not available and were substituted with 24 or 96 h studies. For each pesticide the typical mode of application and mode of action of the pesticide was used to determine whether the primary route of exposure of bees was likely to be via contact or consumption, and appropriate $LD_{50}$ values were then used. For 66 of the most obscure pesticides no $LD_{50}$ values were publicly available. Thirty-six of these were members of chemical groups for which $LD_{50}$ values were available for closely related compounds, and for these the mean $LD_{50}$ value for other members of the chemical group was substituted. The remaining 30 chemicals were excluded from further analysis. Together they accounted for considerably less than 1% of all pesticides used by weight. We then calculated the number of honey bee $LD_{50}$ doses applied each year for each chemical, by dividing the mass applied by the $LD_{50}$. This indicates the potential kill due to use of the chemical, assuming all of it was consumed by or came into contact with honeybees. It is important to reiterate that this is not an attempt to estimate actual bee deaths, for we do not know the proportion of each chemical that will come into contact with bees or other non-target insects.

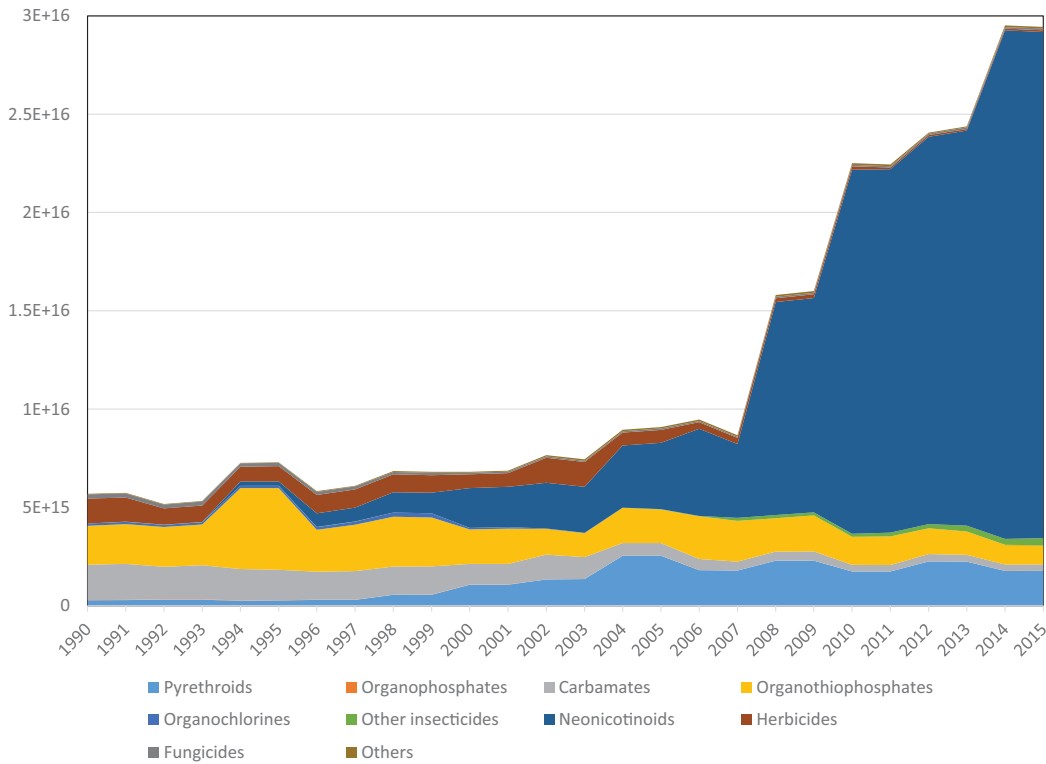

**Figure 2 Potential number of honey bee LD$_{50}$s in pesticides applied to Great British farmland each year.**

## RESULTS

Between 1990 and 2015 the total weight of pesticides used in Great Britain fell by 48% from 34.4 to 17.8 thousand tons per year (Fig. 1). In contrast, the area treated almost doubled, from 45 to 80 million hectares. The area of cropped land has remained approximately stable throughout this period at around 4.6 million hectares (*Department for Environment, Food and Rural Affairs (Defra), 2016*). Thus in 1990 each hectare of cropped land on average received a total of 7.5 kg of pesticide active ingredient delivered in 9.8 applications. By 2015 each hectare of land received 3.9 kg of pesticide in 17.4 applications, a marked change in practice (note that several active ingredients may be applied at once, so this does not mean farmers actually spray fields 17.4 times). In summary, in 2015 each field was being sprayed almost twice as frequently compared to 1990, but with a total mass of pesticide that had nearly halved over the period.

The potential number of bees killed by these applications (the number of LD$_{50}$ doses that could be delivered) rose approximately six-fold over the 26-year period, from $5 \times 10^{15}$ to $30 \times 10^{15}$ (Fig. 2). Toxicity due to herbicides declined over time, largely due to decreased usage of triazines such as simazine, while declines in use of carbamate and organothiophosphate insecticides also reducing the toxic load. However, these reductions were more than offset by increases in toxicity due to an approximately five-fold increase in the weight of pyrethroid insecticides applied, and a very large increase in toxicity due to the introduction and widespread adoption of neonicotinoid insecticides from

1994 onwards. Eighty-seven percent of the total toxic load in 2015 was due to neonicotinoids, and >99% of this was due to three compounds: imidacloprid, clothianidin and thiamethoxam. All three compounds have become widely used and have very low $LD_{50}$ values, in the region of four to five ng per bee via oral exposure.

## DISCUSSION

These data on the number of potential $LD_{50}$ doses must be interpreted with considerable caution. A volume of $30 \times 10^{15}$ is enough to give 10,000 lethal doses to each of the approximately three trillion honeybees in the world. In reality, the very large majority of the pesticides applied will not come in to contact with any bee. The total toxic load entering the environment is just part of the story. The probability of a pesticide coming into contact with a bee will depend on many factors, such as how and when it is applied, what crops it is applied to, its persistence, whether it acts systemically in plants (and hence enters nectar and pollen) and so on. If neonicotinoids were being used in place of more persistent chemicals, or those that were more likely to find their way into nectar and pollen, then this might offset their higher toxicity. However, the opposite appears to be the case; neonicotinoids are persistent in the environment, and being systemic are regularly found in the pollen and nectar of both flowering crops and also wildflowers in farmland (*Krupke et al., 2012*; *Bonmatin et al., 2015*; *Botías et al., 2015*, *2017*; *Mogren & Lundgren, 2016*). As a result, they are often the most common pesticides found in honey and pollen stores in honey bee and bumblebee colonies (*Lambert et al., 2013*; *Sanchez-Bayo & Goka, 2014*; *David et al., 2015*, *2016*), and in wild bees themselves (*Hladick, Vandever & Smalling, 2016*). Indeed, a recent study found neonicotinoids in 75% of honey samples collected from diverse locations around the globe and including remote Pacific islands, suggesting that honey bees do routinely come into contact with these chemicals (*Mitchell et al., 2017*).

Although it is clear that bees are chronically exposed to pesticides, most of the time they are likely to receive sublethal doses. Our approach does not directly capture such effects, but if we make the reasonable assumption that, for each chemical, the doses causing sublethal effects are lower but proportional to the doses causing acute mortality, then our analysis should indicate the likely changes over time in the relative frequency with which bees receive a dose that does them sublethal harm.

We focus here on honey bees for the reason that $LD_{50}$ values are available for honey bees and they are of course major pollinators, but neonicotinoids are highly toxic to all insects that have been tested, both pests and beneficials (*Pisa et al., 2015*). More broadly, toxicity of pesticides to insects tends to be broadly similar across insect species. It is thus likely that all non-target insects including other pollinators are likely to be similarly at risk, which may explain the apparent links between patterns of pesticide use and declines of aquatic insects, butterflies, and insect-eating birds (*Van Dijk, Van Staalduinen & Van der Sluijs, 2013*; *Gilburn et al., 2015*; *Forister et al., 2016*; *Hallmann et al., 2014*). It is also noteworthy that this six-fold increase in potential toxicity to insects in the period 1990–2015 corresponds closely with the timing of the

76% decline in flying insect biomass recorded in Germany in the period 1989–2014 (*Hallmann et al., 2017*).

In conclusion, while acknowledging that our analysis makes many simplifying assumptions, nonetheless it suggests that the risks that pesticides pose to bees and other beneficial insects may have considerably increased in the last 26 years in Great Britain, despite a complex regulatory system and a push from the EU for reduced pesticide use and a move towards IPM. However, the EU recently decided to implement a ban on all field uses of the three neonicotinoids imidacloprid, clothianidin, and thiamethoxam from the end of 2018. We might thus expect a drop in this measure of toxic load in EU states from 2019, perhaps depending on what new pesticides are introduced to replace them.

## ACKNOWLEDGEMENTS

We are very grateful to the Pesticide Usage Statistics team at the Farmed Environment Research Agency, who conduct the pesticide use surveys and make the data freely available.

### Funding

The authors received no funding for this work.

### Competing Interests

The authors declare that they have no competing interests.

### Author Contributions

- Dave Goulson conceived and designed the experiments, analyzed the data, contributed reagents/materials/analysis tools, prepared figures and/or tables, authored or reviewed drafts of the paper, approved the final draft.
- Jack Thompson performed the experiments, analyzed the data, contributed reagents/materials/analysis tools, authored or reviewed drafts of the paper, approved the final draft.
- Amy Croombs performed the experiments, analyzed the data, contributed reagents/materials/analysis tools, authored or reviewed drafts of the paper, approved the final draft.

### Data Availability

Our data are all freely available from https://secure.fera.defra.gov.uk/pusstats.

The raw data for each pesticide can be obtained by clicking "tables" at https://secure.fera.defra.gov.uk/pusstats/myindex.cfm, and then selecting each pesticide in turn using the "active substance" tab.

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
