# Peer review of "Rapid rise in toxic load for bees revealed by analysis of pesticide use in Great Britain"

_PeerJ, doi:10.7717/peerj.5255_

## Round 0.1 · original submission · Major Revisions

The reviewers, who differed markedly in their assessment of your paper strongly question many of the assumptions made in your paper, and I generally support their concerns. However, the paper is on an important topic, and has merit, and therefore I am prepared to consider a major rewrite, focusing on introducing greater rigour in your arguments. I will be closely checking your response to the reviewers, which should address all of their concerns directly and explicitly. In particular, I would ask you to think carefully about many of the extrapolated numbers quoted in your paper, to ensure both that they are supported by evidence, and that the assumptions implicit in calculations are made clear to the reader.

·

Basic reporting

Abstract
So the 3 x 10^6 bees dead is somewhat inflammatory, however you do qualify it with the line detailing the many factors that this number does not take into account. However, following line ’The total potential kill of honeybees (the total number of LD50 doses applied to the 4.6 million….’ I would like you to have some further caveat in the form of ‘While this does not mean this number of bees would be killed, it provides a n indication of the relative change in scale of pesticide application over this period’.

Introduction
L55: You mention this above, but maybe make it clear that while the paper is orientated towards risks for bees exposure to pesticides poses a threat for numerous beneficial (pest control, decomposition) and biodiversity components of the agro-ecosystem. Given what you are doing your findings would seem to be relevant to all of these groups, and in fact bees are more of a special case in the sense that a lot of the time the impact is linked to mass flowering crops (although not exclusively)
L57-58: I don’t think that’s true. I had a feeling that either Holland or Denmark had very detailed pesticide records (although possibly not going back that far). I think its reasonable though to use GB as a model to asses this issue given its good records and intensive arable agriculture.
L62-64: Really all this does is scale it relative to honeybees, although arguably it could be done for any invertebrate used as a model system in the regulatory process (although presumably you would get slightly different results depending on species specific sensitivities). I am not asking you to change this to include other species, but maybe you could make that point.
L62-64: Another point here is that (and you were getting that point across to an extent in the abstract) is that this is to provide an idea of the scale of increase, rather than a quantification of mortality. Specifically LD50 relate to a more or less instantaneous dose. My point being is that as long as that dose on a given field for a given application is less than the LD50 from a regulatory perspective (more or less) that’s not a major problem. By adding over the course of a year you may make the problem seem worse that it is. As long as you make it clear this is in a way a ‘unit’ of comparison between years then it should be ok – but it does need to be explicitly explained. You may want to justify this by stating that simply considering pesticides in terms of kg of active ingredient applied per unit area does not give any idea of relative toxicity of compounds comparable over time from 1990 to 20??
L67 extra .
Methods
L79 state (oral or contact Ld50 values)
L86-87: Again qualify this and say it s unlikely, but it provides a useful measure for comparison between years for compounds that differ in toxicity…or something like that.

Results
L96. Why the square brackets?

Discussion
Short and to the point…no real problems here.

Experimental design

see above

Validity of the findings

see above

Additional comments

I like this paper, I think it has an important message about the relative increase in pesticide use. I think by focussing on LD50s and using these to scale relative pesticide toxicities between years (1990’s into the 2010’s) it provides an interesting relative metric that would be obscured by the use of simply tonnes of a.i. that said caveats need to be clearly defined that the values given do not necessarily mean 10^15 bees would have been killed (these I highlight below). As long as that can be made clear (and I see no reason why not) this is an interesting paper. The caveats need to be in there and to be clearly stated to stop misinterpretation by the press, although I think you will be lucky not to have this interpreted as pesticides kill 10^15 bees.

Reviewer 2 ·

Basic reporting

The reporting is clear, the English good.
The literature references are some what one-sided, although many recent ones included. Why not refer to the well balanced Restatements of Godfray et al. (2014, 2015).? A majority of citations is to members of the own group of (co-) authors, alsdo in cases where other reviews and data compilations are available.

Experimental design

The research question is well defined, but I doubt the relevance as well as its meaning. Actually it is more covering what is really going on than revealing. This is the choice of the authors.

Validity of the findings

Data is OK, but is used to SUGGEST a conclusion, or actually a meaning. One could argue that this is not speculation, and indeed the authors acknowledge that preservation (lines 64-66; lines 86-87; lines 111-118), nevertheless they step over it and use the evidence just to SUGGEST that the risk has increased a six fold. Which cannot be derived from this analysis.

Additional comments

I will give some general and specific comments here:
The paper is generous with wordings alike: 'suggest', seem, appear, which does not improve the scientific clarity but serves a more political aim, the reader has to become aware that the potential kill has increased (despite the preservation this maybe only ssemingly the case). Actually the aim of the paper: (Lines 55-56) to evaluate the RISKS to bees ... cannot be reached since only the 'poterntial kill says hardly anything (thus nothing0 about risks.
L 30: there is broad consensus: why hide behind the authority of other scinentists, but only cite a own paper? Note that consensus is very often not a good 'quality' criterium.
L49-52: correctly the Authors state that 'Recent studies SUGGEST that ......is CONTRIBUTING....., since most of the studies just show correlations, the Van Dijk study did ignore correlations with other factors (see Vijver & van den Brink, 2014), or were heavily disputed (the Hallamann paper on birds). So 'correlate with' is considered similar to 'suggest to contribute'.
L 58: there are many regions with data on pesticide use in Europe, so I wonder if the statement is true.
L73-74: assuming the same impact of 10 times concentration 1 to 1 time concentration 10 is really nonsens (many species can detoxify low concentrations), only in case of time reinforced toxicity this assumption might be valid (but needs not to be).
L 81-83: it is tricky to derive an LD50 when unknown from 'related' compounds (compare f.i. the Neonicotinoids imidacloprid and thiacloprid! Happily it did not account for a high percentage of cases (L 84).
L104-105: due to the neonics the toxic load APPLIED increased-but how much did the toxic load of the actual exposure increase? Or did it decrease? Neoinics are used primarily in seed coatings, by which only a very small share shows up in the plant (of which a small part in honey and pollen).
L120-128: that neonicotinoids are found in honey, and several other pesticides not, does not reveal anything about their exposure or risk, but just reflects their hydrophilic properties (by which they can systemically withgin the plant reach the pollen and nectar).
L 130-131: this is no reasonable assumption, unless well argued, or supported by data
L 146: assumption should be assumptionS

Conclusion: the analysis SUGGESTS (citation): My question: does this SUGGESTION, based on an analysis with 'many simplifying assumptions' (citation), warrant a scientific publication?

Reviewer 3 ·

Basic reporting

The language is at times jocular and detracts from the potential impact of the narrative. E.g., Ln 114 "if this were so, there would be no bees left in Britain".

Experimental design

The results section needs further development it is insufficiently descriptive and the values given cannot be adequately assessed as presented.

Validity of the findings

Data as presented are not sufficient for the treatment of the material as presented.

Additional comments

This manuscript addresses an important and controversial topic, namely the link between pesticide usage rates and bee declines in Great Britain. However, the data presented are not sufficiently convincing to support the linking of these two narratives at this juncture. For instance, the values presented (e.g., Ln 90-97) inadequately capture the complexity of the changes in pesticide usage and treated areas. Specifically, is the amount of pesticide given the ‘purchased’ rather than ‘applied’ quantity? Further, how has the cropped land has remained stable and what areas/sectors are being sprayed if the (Ln 91) ‘area treated almost doubled’? At a minimum, further description is needed in the results section to increase clarity and impact. Also, the reasoning behind the development of a worst-case scenario, or ‘comparative measure’ (e.g., Ln 65) should be further developed. For instance, the amount of pesticides found in honey and pollen stores mentioned in the discussion (Ln 120-128) could be moved to the introduction to lend support to the reasoning behind this approach from the outset. Finally, the discussion would also benefit from the treatment of other likely contributing stressors and / or the contribution from excluded ‘obscure’ pesticides.

---

## Round 0.2 · accepted · Accept

The reviews of your original article submission were mixed, and in requesting a major revision, I indicated that I would pay close attention to how you addressed the criticisms of the reviewers, which although divergent in their opinions of the paper, all agreed on the point that you should provide additional evidence and explanatory text to support claims regarding the numbers of bee deaths directly arising from neonicotinoid exposure. Given the ongoing controversy around the use and potential impacts of neonicotinoids, particularly as related to pollinators, it is expedient for any researcher publishing new findings to ensure that readers can fully understand the assumptions underlying any statements of potential historical or future impacts. In particular, your estimates of potential bee deaths associated with neonicotinoid use may not represent actual bee deaths, and are likely to be strongly challenged. However, I am pleased that you have taken the necessary steps to make assumptions explicit, and to clearly state in the paper that this is an estimate based on circumstances which are likely to be highly variable in space and time. I am satisfied that you have addressed the reviewers’ criticisms comprehensively.